# Photoperiodic Requirements for Induction and Maintenance of Rhythm Bifurcation and Extraordinary Entrainment in Male Mice

**Jonathan Sun**[ID]**, Deborah A. M. Joye †, Andrew H. Farkas ‡ and Michael R. Gorman ***[ID]

Department of Psychology and Center for Circadian Biology, University of California, San Diego, CA 92093, USA
* Correspondence: mgorman@ucsd.edu; Tel.: +1-858-822-2466
† Current address: Department of Biomedical Sciences, Marquette University, Milwaukee, WI 53233, USA.
‡ Current address: Department of Psychology, University of Georgia, Athens, GA 30602, USA.

**Abstract:** Exposure of mice to a 24 h light:dark:light:dark (LDLD) cycle with dimly illuminated nights induces the circadian timing system to program two intervals of activity and two intervals of rest per 24 h cycle and subsequently allows entrainment to a variety of extraordinary light regimens including 30 h LDLD cycles. Little is known about critical lighting requirements to induce and maintain this non-standard entrainment pattern, termed "bifurcation," and to enhance the range of apparent entrainment. The current study determined the necessary duration of the photophase for animals to bifurcate and assessed whether requirements for maintenance differed from those for induction. An objective index of bifurcated entrainment varied with length of the photophase over 4–10 h durations, with highest values at 8 h. To assess photic requirements for the maintenance of bifurcation, mice from each group were subsequently exposed to the LDLD cycle with 4 h photophases. While insufficient to induce bifurcation, this photoperiod maintained bifurcation in mice transferred from inductive LDLD cycles. Entrainment to 30 h LDLD cycles also varied with photoperiod duration. These studies characterize non-invasive tools that reveal latent flexibility in the circadian control of rest/activity cycles with important translational potential for addressing needs of human shift-workers.

**Keywords:** circadian entrainment; T cycles; bifurcation; phase-shifting; shift-work; SCN; oscillator coupling

## 1. Introduction

Because the earth's rotation generates daily variations in environmental conditions, organisms have evolved internal circadian timing systems to temporally organize physiological and behavioral processes. In mammals, a neural pacemaker in the suprachiasmatic nuclei (SCN) of the anterior hypothalamus maintains a robust internal representation of daily time and coordinates synchrony with the environmental light cycle via afferents from specialized photoreceptive ganglion cells in the retina [1]. Cellular rhythmicity in the SCN and other tissues depends on a transcription–translation feedback loop in individual cells that involves a handful of core clock genes, their protein products, post-translational modification, and degradation processes [2]. Isolated cells of the SCN express a wide range of circadian periods, but can coordinate via intercellular coupling mechanisms to generate a robust, stable, network-level rhythm [3]. The SCN network directly and indirectly entrains cellular oscillators throughout the body via a variety of secretory, synaptic, thermal, and metabolic coupling mechanisms [3,4]. Although the light/dark cycles are pervasive, they are not constant. Sun-dialing enthusiasts are well aware that the earth's rotational period relative to the solar sun deviates over the course of the year, requiring significant correction by the so-called equation of time. Of greater

biological relevance, at all points apart from the equator the axial tilt of the earth introduces seasonal variation in the relative durations of day and night. Consequently, animals that experience seasonal variation in day length presumably have strong selection pressures for their circadian clocks to adapt to the changing waveform of the day. Formal and physiological mechanisms of photoperiodic plasticity have been studied extensively in diverse species [5–8].

Regardless of the organism under study, functional plasticity in clock function is particularly evident in the form of after-effects, in which clock parameters measured under identical conditions vary as a function of the conditions to which the clock was previously exposed. In many species, free-running period in constant darkness is longer or shorter depending on whether subjects were entrained to photoperiods that were longer or shorter than 24 h, respectively (i.e., period after-effects) [9,10]. Furthermore, after entrainment to long summer-like versus short winter-like photoperiods, the waveforms of rhythms in locomotor activity, nightly melatonin secretion, and SCN function remain distinctly different for days or weeks under constant conditions (i.e., waveform after-effects) [11,12]. After-effects can also be detected under light/dark conditions: Exposure of Siberian hamsters to daylengths above a critical photoperiod prevents some individuals from entraining to winter photoperiods with a seasonally appropriate waveform [13]. In the same species, exposure to a particular set of phase shifts also renders animals arrhythmic and un-entrainable (i.e., entrainment after-effects) to any tested condition, despite no evident neuropathology [14]. In each of these examples, the central pacemaker appears to be enduringly altered as a function of the photoperiodic conditions to which the clock was exposed.

An interesting feature of waveform after-effects is that they are associated with altered responsiveness to phase-resetting and/or entraining actions of light. The phase response curve (PRC) to bright pulses of light, for example, shifts from weak, Type 1, resetting in animals with summer waveforms to stronger, sometimes Type 0, resetting in animals with winter waveforms [15–17]. Apart from the size of the phase shift elicited by bright light pulses, the sensitivity to light—assayed by how much light is required to elicit half of the maximum phase shift—is increased forty-fold in short versus long daylength hamsters [18,19]. A distinct waveform manipulation, termed bifurcation, permits stable entrainment to a range of non-24 h periods that have long been considered outside the range of entrainment for rodents. Bifurcation refers to the division of subjective night and subjective day into two components each in response to permissive 24 h light:dark:light:dark (LDLD) conditions [20]. The term is used descriptively and does not imply any relation to usages in mathematical analyses of dynamical systems or other contexts. Following entrainment bifurcation, mice and hamsters are capable of exhibiting apparent entrainment to LDLD cycles as short as 18 h and as long as 30 h [21–23]. For this and other reasons, waveform manipulations such as rhythm bifurcation may have translation utility for adapting human clocks to the demands of shift-work [24].

While bifurcation has been the subject of intensive study in this laboratory, the specific requirements for its induction and maintenance have yet to be rigorously defined. In each species examined, bifurcation is markedly facilitated by exposure to weak nighttime lighting equivalent in intensity to that of bright starlight/dim moonlight (e.g., <0.1 lux), as opposed to physiological darkness [21,25,26]. Other factors that encourage bifurcation in mice include availability of a running wheel, brighter photophases, younger age, and female sex [27]. In the present studies, we determined that the relative duration of light and dark differentially influences induction and maintenance of bifurcation in 24 h LDLD cycles. Additionally, we demonstrate that 24 h bifurcation permits subsequent entrainment to 30 h LDLD cycles, again with stricter photoperiod requirements for induction than for maintenance of this enhanced behavioral entrainment.

## 2. Materials and Methods

### 2.1. General Methods

Male C57Bl/6J mice were purchased at 5 weeks from Jackson Labs (Sacramento, CA, USA; Experiment 1) or bred in the laboratory from Jackson stock (Experiment 2), provided with food (Mouse Diet 5015; Purina, St. Louis, MO, USA) and water ad libitum. Mice were 4–8 weeks of age at the beginning of the experiments. Experimental cages were housed in light-tight ventilated chambers holding up to 16 cages each. Light in the photophases was generated by white fluorescent lamps providing illumination of 30–100 lux inside individual cages. No nighttime lighting was provided to the animal colony, but once experimentation began, all animals were exposed to dim scotopic illumination from narrow-band LEDs (peak and half max bandwidth = 561 and 23 nm, respectively) at an intensity of <0.1 lux in the brightest positions in the cage. All procedures were reviewed and approved by the UCSD Institutional Animal Care and Use Committee (Protocol Number: S00061M, 2000).

### 2.2. Experimental Procedures

**Experiment 1.** *Photoperiod Dependence of Bifurcation Induction and Maintenance in 24 h T LDLD Cycles.*

Mice (*n* = 40) were transferred from same-sex group housing in the colony to single-housing in wheel-running cages (~29 × 19 × 17 cm, 12 cm diameter). After 10 days in $LD_{im}$ 14:10 (Phase 0), mice were exposed to one of four 24 h LDLD conditions for 4 weeks (Phase 1; Table 1). The four conditions varied in terms of the relative lengths of the photophases. As the 24 h LDLD (T24) conditions used in these experiments are all equivalent to 12 h LD cycles (T12), they are represented here in their simpler form: $LD_{im}$ 4:8 (*n* = 16); $LD_{im}$ 6:6 (*n* = 8); $LD_{im}$ 8:4 (*n* = 8) and $LD_{im}$ 10:2 (*n* = 8). Subsequently, all groups were exposed to $LD_{im}$ 4:8 for 6 weeks to determine whether there were after-effects of prior entrainment (Phase 2; Table 1). Throughout, the onset of one scotophase was unchanged from that in the colony, and changes in photoperiod were accomplished by altering the time of light onset. In Phase 3, all mice were exposed for an additional 2 weeks to constant dim illumination ($D_{im}D_{im}$) beginning during the scotophase anti-phase to the original colony scotophase (Table 1).

**Table 1.** Experimental lighting conditions for Experiments 1 and 2.

| Experiment 1 | # Cycles @ T | Photoperiod Conditions | | | |
|---|---|---|---|---|---|
| Precondition | 10 @ 24 h | $LD_{ark}$ 14:10 (*n* = 40) | | | |
| Phase 1—Bifurcation Induction | 56 @ 12 h | $LD_{im}$ 4:8 (*n* = 16) | $LD_{im}$ 6:6 (*n* = 8) | $LD_{im}$ 8:4 (*n* = 8) | $LD_{im}$ 10:2 (*n* = 8) |
| Phase 2—Bifurcation After-effects | 84 @ 12 h | $LD_{im}$ 4:8 | $LD_{im}$ 4:8 | $LD_{im}$ 4:8 | $LD_{im}$ 4:8 |
| Phase 3—Constant Conditions | 14 @ 24 h | $D_{im}D_{im}$ | $D_{im}D_{im}$ | $D_{im}D_{im}$ | $D_{im}D_{im}$ |
| Experiment 2 | # Cycles @ T | Photoperiod Conditions | | | |
| Precondition | 10 @ 24 h | $LD_{ark}$ 14:10 (*n* = 44) | | | |
| Phase 1a/1b—Bifurcation Induction | 56 @ 12 h | $LD_{im}$ 4:8 (*n* = 10) | $LD_{im}$ 5:7 (*n* = 12) | $LD_{im}$ 6:6 (*n* = 12) | $LD_{im}$ 7:5 (*n* = 10) |
| Phase 1c—Dark nights | 28 @ 12 h | $LD_{ark}$ 4:8 | $LD_{ark}$ 5:7 | $LD_{ark}$ 6:6 | $LD_{ark}$ 7:5 |
| Phase 1d—Dim nights | 28 @ 12 h | $LD_{im}$ 4:8 | $LD_{im}$ 5:7 | $LD_{im}$ 6:6 | $LD_{im}$ 7:5 |
| Phase 2—T15 Entrainment | 26 @ 15 h | $LD_{im}$ 7:8 | $LD_{im}$ 8:7 | $LD_{im}$ 9:6 | $LD_{im}$ 10:5 |
| Phase 3—T15 After-effects | 56 @ 15 h | $LD_{im}$ 7:8 | $LD_{im}$ 7:8 | $LD_{im}$ 7:8 | $LD_{im}$ 7:8 |

**Experiment 2.** *Photoperiod Dependence of T24 Bifurcation Induction and Entrainment to T30 LDLD Cycles.*

Mice (*n* = 44) were treated as described above through Phase 1 of Experiment 1 except that the Phase 1 conditions to which they were assigned were $LD_{im}$4:8 (*n* = 10), $LD_{im}$5:7 (*n* = 12), $LD_{im}$6:6 (*n* = 12) and $LD_{im}$7:5 (*n* = 10; Table 1). After 4 weeks, the dim nocturnal illumination was extinguished for two weeks and then restored for an additional two weeks (Table 1). To assess photoperiodic dependence of entrainment to extreme non-24 h days, all mice were transferred to T30 LDLD (equivalent to T15 LD) for approximately 2 weeks (Phase 2) by extending the duration of each light phase by 3 h (i.e., $LD_{im}$7:8, $LD_{im}$8:7, $LD_{im}$9:6 and $LD_{im}$10:5; Table 1). To examine possible after-effects of entrainment to different T30 photoperiods, mice from all groups were exposed to $LD_{im}$7:8 for 4 weeks (Phase 3; Table 1).

*2.3. Data Collection and Analyses*

Activity rhythms were monitored with VitalView software (Version 4.2, Mini-Mitter, Bend, OR, USA), which counts half wheel revolutions recorded in 6-min bins. Cage changes for all animals occurred at approximately three-week intervals and were conducted at least 2 h into the photophase to minimize circadian disruption. In the event that a cage change was scheduled the same week as a light manipulation, the cage change was either postponed or advanced to minimize disturbance.

2.3.1. Bifurcation Symmetry Index (BSI)

As described previously [21], we devised a simple continuous metric to objectively assess the degree to which LDLD cycles resulted in division of activity between alternate (i.e., anti-phase) scotophases. Accordingly, in each 24 period, we determined the number of wheel-running revolutions in each individual scotophase (Scotophase1, Scotophase2) and the total amount of activity over the 24 h cycle (Total LDLD activity). For each day, the lesser of the two scotophase values was divided by the total activity and doubled: min(Scotophase1, Scotophase2)*2/Total LDLD activity. These daily values were averaged for each animal over 12–14 cycles to yield a BSI. If activity is exactly divided between scotophases, each will have 50% of the daily average and BSI = 1. If activity is exclusive to one scotophase, BSI = 0. If activity is unevenly divided, or if some activity occurs in the light (as occurs when rhythms are free-running), BSI values will be intermediate. A comparable calculation over 30 h LDLD intervals was used to calculate BSI in T15 conditions.

2.3.2. Entrainment Quotient in T15/T30

We devised another metric to objectively quantify how well activity rhythms entrained to 30 h LDLD cycles [22]. Lomb–Scargle periodograms, which are generated by fitting cosine functions to time series data, were calculated over 14 days (~336 h) intervals (ClockLab, Actimetrics, Evanston, IL, USA). Periodogram power at 15 h and at 30 h was noted. If robust and similar activity bouts occurred in alternate scotophases, then statistical power would be high at 15 h, but there would be no power at 30 h because alternate activity bouts would fall in anti-phase of a 30 h cycle. If activity was present in alternate scotophases but the amount or patterning of activity different systematically, then periodogram power would be expected to be elevated at both 15 and 30 h. Generally, T30 power was minimal. We additionally estimated the strength of rhythmicity not entrained by the T15/30 zeitgeber by noting and taking account of the peak periodogram power in the circadian range (i.e., 23–26 h). Rhythmicity at any other values or range was negligible. These multiple periodogram values were integrated into a single measure, an entrainment quotient (EQ), by summing values matching the zeitgeber (periodogram power at 15 h + power at 30 h) and dividing by the sum of periodogram power in the zeitgeber and circadian range (periodogram power at 15 + power at 30 h + peak power 23–26 h). EQ approached 1 when all power matched the zeitgeber (whether 15 h or 30 h) and no residual circadian rhythmicity could be determined. EQ approached 0 when only a non-entrained circadian rhythm was observed. A free-running rhythm with negative or positive masking by light would generate intermediate EQ values.

### 2.3.3. $D_{im}D_{im}$ Data

Under constant dim conditions, free-running period (FRP) was estimated as the value generating the greatest statistical power in the Lomb–Scargle periodogram (Periodogram FRP). Additionally, FRP was estimated as the slope of least-squares regression lines to eye-fit edited activity onsets in ClockLab (Onset FRP, Actimetrics; Wilmette, IL, USA)**.** From eye-fit activity onsets, the projected phase of activity onset on the day of transfer to $D_{im}D_{im}$ was also estimated.

### 2.3.4. Statistical Tests

Because data in some cases violated assumptions of parametric statistical tests, we compared group means using the Kruskal–Wallis test, a non-parametric equivalent of ANOVA. Where the test-statistic H, which approximates a chi-square distribution, indicated that the null hypothesis of no differences between groups should be rejected, pairwise group comparisons were performed using the non-parametric Wilcoxon test. Clustering of activity onsets was assessed with Rayleigh tests. As a result of poor health, one animal was removed from Experiment 1 (Group $LD_{im}10:2$) and one from Experiment 2 (Group $LD_{im}5:7$).

## 3. Results

### 3.1. Experiment 1

Results are presented first descriptively based on illustrative wheel-running actograms of individual mice from each group (Figure 1) and then quantitatively based on objective group averages (Table 2). The induction of bifurcated activity rhythms in Phase 1 would be reflected in top third of actograms (Figure 1) as the steady state appearance of robust locomotor activity in each of the two scotophases per 24 h cycle. The mouse in Figure 1A exemplifies the pattern of 14 of 16 mice exposed to $LD_{im}4:8$. Activity is concentrated in only one of the two daily scotophases. Subjectively, the rhythm is clearly unbifurcated, and this subjective judgment is corroborated by the low BSI value of 0.04. Only two of 16 mice in this group expressed robust locomotor activity in both scotophases. In both cases, the bimodal activity rhythms were not stable (data not shown). In $LD_{im}8:4$, by contrast, the selected actogram (Figure 1C, BSI = 0.88) features activity that is nearly evenly distributed between the two photophases, a pattern that characterized every mouse in that condition. In $LD_{im}6:6$ and $LD_{im}10:2$, a greater variety of patterns was seen between individual mice. Both of the examples illustrate divided activity between scotophases (Figure 1B,D) but also evidence unstable or poorly entrained activity components.

Quantification of group entrainment parameters revealed that BSI in Phase 1 was highest in $LD_{im}8:4$ and lowest in $LD_{im}4:8$ mice (Table 2, Figure 2A). BSI of mice in $LD_{im}10:2$ and $LD_{im}6:6$ were both significantly elevated relative to $LD_{im}4:8$. These groups did not differ from each other, but both were significantly lower than in $LD_{im}8:4$. This same pattern of group differences was mirrored by the computationally-independent analyses of Lomb–Scargle periodogram power at 12 and 24 h, except 12 h periodogram power also differed between $LD_{im}6:6$ and $LD_{im}10:2$ mice (Table 2).

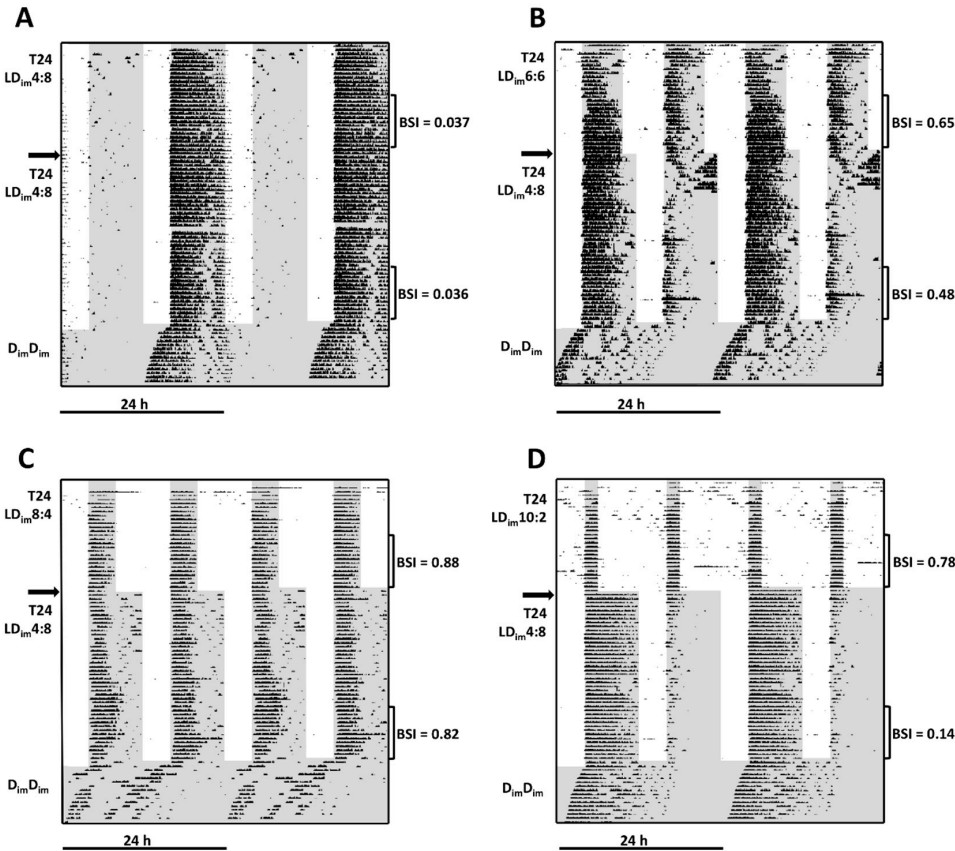

**Figure 1.** Illustrative wheel-running actograms of mice from each condition in Experiment 1. Actograms (**A–D**) are double-plotted modulo 24 h and scaled from 0 to 100 wheel revolutions per minute compiled in 6 min bins. Mice were initially exposed to $LD_{im}4{:}8$ (**A**), $LD_{im}6{:}6$ (**B**), $LD_{im}8{:}4$ (**C**) or $LD_{im}10{:}2$ (**D**) for 4 weeks (upper third of actograms) followed by 6 weeks of exposure to $LD_{im}4{:}8$ (middle of actograms) and, finally, 2 weeks of $D_{im}D_{im}$ (lower portion of actograms). Scotophases are indicated with dark shading. BSI values are shown for these individual animals and indicate the intervals analyzed for all animals to generate data averaged in Table 2.

**Table 2.** Group mean ± sem of circadian parameters measured in Experiment 1.

| | Group | | | | Kruskal–Wallis Test |
|---|---|---|---|---|---|
| **Phase 1 (T24 Photoperiods)** | $LD_{im}4{:}8$ | $LD_{im}6{:}6$ | $LD_{im}8{:}4$ | $LD_{im}10{:}2$ | |
| BSI | 0.17 ± 0.04 A | 0.59 ± 0.09 B | 0.93 ± 0.01 C | 0.69 ± 0.05 B | H(3) = 28.9; $p < 0.001$ |
| Periodogram-12 h | 349 ± 28 A | 691 ± 58 B | 869 ± 51 C | 461 ± 28 D | H(3) = 28.7; $p < 0.001$ |
| Periodogram-24 h | 633 ± 50 A | 208 ± 68 B | 3 ± 1 C | 13 ± 3 B | H(3) = 30.3; $p < 0.001$ |
| **Phase 2 (T24 After-effects)** | $LD_{im}4{:}8$ | $LD_{im}4{:}8{:}8$ | $LD_{im}4{:}8$ | $LD_{im}4{:}8$ | |
| BSI | 0.08 ± 0.02 A | 0.33 ± 0.09 B | 0.61 ± 0.10 B | 0.42 ± 0.08 B | H(3) = 18.7; $p < 0.001$ |
| Periodogram-12 h | 235 ± 19 | 318 ± 66 | 418 ± 70 | 264 ± 58 | H(3) = 4.3; ns |
| Periodogram-24 h | 646 ± 53 A | 372 ± 87 B | 169 ± 101 B | 224 ± 83 B | H(3) = 18.0; $p < 0.001$ |
| **Phase 3 (Constant dim)** | $D_{im}D_{im}$ | $D_{im}D_{im}$ | $D_{im}D_{im}$ | $D_{im}D_{im}$ | |
| Onset FRP (h) | 23.68 ± 0.03 A | 23.71 ± 0.05 A | 23.45 ± 0.09 B | 23.81 ± 0.03 A | H(3) = 11.7; $p < 0.01$ |
| Periodogram FRP (h) | 23.70 ± 0.04 | 23.70 ± 0.08 | 23.58 ± 0.08 | 23.73 ± 0.04 | H(3) = 2.7; ns |
| Peak periodogram power | 553 ± 68 A | 341 ± 56 AB | 369 ± 98 AB | 229 ± 69 B | H(3) = 8.9; $p < 0.05$ |
| *n* | 16 | 8 | 8 | 7 | |

Groups with non-overlapping letters differ significantly by Wilcoxon test

After exposure of each experimental group to LD$_{im}$4:8 in Phase 2, mice generally, but not always, maintained their bifurcation status from Phase 1. As expected, entrainment is essentially unchanged across Phases 1 and 2 in the representative LD$_{im}$4:8 mice (Figure 1A sections above and below the left horizontal arrow), where lighting conditions were static. Among the subjects initially exposed to LD$_{im}$8:4 (Figure 1C), in contrast, a clear bifurcated rhythm is evident in LD$_{im}$4:8 (BSI = 0.82). In other groups, transfer to LD$_{im}$4:8 sometimes coincided with rhythm reorganization: the selected actogram (Figure 1B) from LD$_{im}$6:6, for example, shows transiently increased activity in the non-dominant scotophase after transfer to LD$_{im}$4:8, but activity stabilizes with a reduce BSI compared to Phase 1. In contrast, following transfer from LD$_{im}$10:2 (Figure 1D), the previously divided rhythm immediately adopts a typically non-bifurcated entrainment pattern with low BSI.

Across groups, BSI decreased in Phase 2 compared to Phase 1 (Table 2; Figure 2A). Objective rhythm analyses, now all calculated from mice in the 5th and 6th week of Phase 2 exposure to LD$_{im}$4:8, illustrate the enduring after-effects of the prior photoperiods (Table 2; Figure 2A). BSI was statistically lower in mice originating in LD$_{im}$4:8 compared to all other groups. Whereas periodogram power at 12 h did not differ by group, power at 24 h was also significantly greatest in mice that were initially exposed to LD$_{im}$4:8 (Table 2).

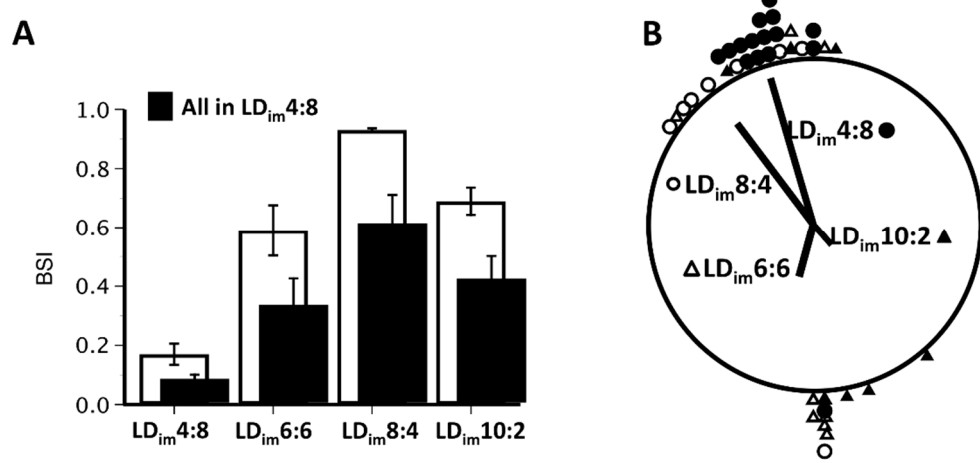

**Figure 2.** Mean ± sem BSI values (**A**) in various Phase 1 photoperiods (open bars) and in Phase 2 LD$_{im}$4:8 (filled bars) of Experiment 1. Individual data points and group mean vectors (**B**) of projected activity onsets relative to the beginning of D$_{im}$D$_{im}$.

In D$_{im}$D$_{im}$ (Phase 3), bifurcated rhythms typically reorganized within several cycles to resemble those of typically unbifurcated subjects (Figure 1C). The period of the free-running rhythm in D$_{im}$D$_{im}$ did not vary significantly between groups as assessed from periodograms but was shorter in LD$_{im}$8:4 mice compared to all others as assessed with activity onsets (Table 2). Peak periodogram power was significantly greater in animals originating in LD$_{im}$4:8 compared to those beginning in LD$_{im}$8:4 (Table 2). Group differences were also apparent in the phase-clustering of free-running activity onsets (Figure 2B). Mice maintained continuously in LD$_{im}$4:8 free-ran from a phase associated with the scotophase that hosted their nocturnal activity. Activity onsets projected to the first cycle in D$_{im}$D$_{im}$ were statistically clustered because all but one of these animals maintained continuous entrainment with activity in the scotophase coincident with the original colony conditions. Despite bifurcation, activity onsets of mice initially entrained to LD$_{im}$8:4 were likewise statistically clustered at approximately the same phase (Figure 2B), whereas one was near anti-phase. Onsets of the mice previously entrained to LD$_{im}$6:6 and LD$_{im}$10:2 were not statistically clustered in D$_{im}$D$_{im}$. Rather, in both cases their distributions tended to cluster around one or the other of the two dark onsets from LDLD rather than to be completely randomly distributed.

## 3.2. Experiment 2

A selected actogram from each group is depicted in Figure 3. As in Experiment 1, mice initially exposed to LD$_{im}$4:8 stably entrained with activity concentrated in the scotophase coinciding with night in original colony conditions. The activity rhythm was not notably changed when dim illumination was discontinued after four weeks and restored two weeks later (Figure 3A). In contrast, at least some animals from each group with longer photophases exhibited bifurcated entrainment (Figure 3B–D), but again without any discernible influence of transitions between dim and dark nights.

Quantitatively, BSI increased monotonically with photophase duration as assessed in both Dark (Phase 1c) and Dim (Phase 1d) scotophase conditions (Table 3). A statistically graded dependence of BSI on photophase duration was evident when assessed under Dark but not Dim scotophases, although in both cases LD4:8 (Dim or Dark) conditions were less inductive of bifurcation than were LD6:6 or LD7:5. Parallel results were observed in computationally-independent Lomb–Scargle periodogram power at 12 and 24 h (Table 3). Values for replicate groups in Experiments 1 and 2 were comparable (Tables 2 and 3).

**Table 3.** Group mean ± sem of circadian parameters measured in Experiment 2.

| | Group | | | | |
|---|---|---|---|---|---|
| **Phase 1c (T24-Photoperiods) *** | **LD$_{ark}$4:8** | **LD$_{ark}$5:7** | **LD$_{ark}$6:6** | **LD$_{ark}$7:5** | **Kruskal–Wallis Test** |
| BSI | 0.25 ± 0.07 A | 0.47 ± 0.07 AB | 0.55 ± 0.05 BC | 0.71 ± 0.05 C | H(3) = 17.1; $p$ < 0.001 |
| Periodogram-12 h | 336 ± 42 A | 594 ± 60 B | 661 ± 66 B | 667 ± 81 B | H(3) = 12.7; $p$ < 0.01 |
| Periodogram-24 h | 451 ± 90 A | 298 ± 70 AB | 173 ± 34 B | 82 ± 27 C | H(3) = 15.2; $p$ < 0.01 |
| **Phase 1d (T24-Photoperiods)** | **LD$_{im}$4:8** | **LD$_{im}$5:7** | **LD$_{im}$6:6** | **LD$_{im}$7:5** | |
| BSI | 0.22 ± 0.04 A | 0.55 ± 0.09 AB | 0.61 ± 0.06 B | 0.70 ± 0.03 B | H(3) = 16.7; $p$ < 0.001 |
| Periodogram-12 h | 271 ± 38 | 382 ± 94 | 410 ± 65 | 412 ± 82 | H(3) = 1.9; ns |
| Periodogram-24 h | 364 ± 76 A | 105 ± 35 B | 107 ± 30 B | 61 ± 14 B | H(3) = 14.5; $p$ < 0.01 |
| **Phase 2 (T30-Photoperiods)** | **LD$_{im}$7:8** | **LD$_{im}$8:7** | **LD$_{im}$9:6** | **LD$_{im}$10:5** | |
| BSI | 0.41 ± 0.05 A | 0.57 ± 0.07 AB | 0.66 ± 0.05 BC | 0.75 ± 0.02 C | H(3) = 17.4; $p$ < 0.001 |
| Periodogram-12 h | 225 ± 34 | 390 ± 99 | 401 ± 75 | 411 ± 72 | H(3) = 3.7; ns |
| Periodogram-24 h | 4.6 ± 2.2 | 3.5 ± 1.0 | 6.8 ± 2.3 | 1.8 ± 0.5 | H(3) = 3.9; ns |
| Periodogram 23-26 h | 175 ± 43 A | 81 ± 34 B | 48 ± 20 B | 21 ± 6 B | H(3) = 16.0; $p$ < 0.01 |
| EQ | 0.59 ± 0.06 A | 0.80 ± 0.07 B | 0.88 ± 0.04 BC | 0.95 ± 0.01 C | H(3) = 18.0; $p$ < 0.001 |
| % activity in light | 2.7 ± 0.4 | 3.6 ± 0.7 | 4.9 ± 1.2 | 3.0 ± 0.5 | H(3) = 1.6; ns |
| **Phase 3 (T30-After-effects)** | **LD$_{im}$7:8** | **LD$_{im}$7:8** | **LD$_{im}$7:8** | **LD$_{im}$7:8** | |
| BSI | 0.55 ± 0.05 | 0.63 ± 0.05 | 0.63 ± 0.05 | 0.72 ± 0.03 | H(3) = 5.8; ns |
| Periodogram-15 h | 119 ± 14 | 205 ± 46 | 145 ± 21 | 109 ± 22 | H(3) = 3.1; ns |
| Periodogram-30 h | 1.6 ± 0.6 | 4.0 ± 2.2 | 1.7 ± 0.8 | 1.6 ± 0.6 | H(3) = 1.0; ns |
| Periodogram 23-26 h | 78 ± 29 A | 43 ± 19 AB | 66 ± 29 AB | 8 ± 2 B | H(3) = 14.1; $p$ < 0.01 |
| EQ | 0.66 ± 0.05 A | 0.80 ± 0.06 AB | 0.78 ± 0.06 AB | 0.91 ± 0.03 B | H(3) = 11.2; $p$ < 0.05 |
| % activity in light | 2.9 ± 0.6 | 3.3 ± 1.0 | 2.8 ± 0.6 | 2.2 ± 0.3 | H(3) = 1.5; ns |
| *n* | 10 | 11 ** | 12 | 10 | |

* Phase 1a and 1b data were similar to subsequent phases and results are omitted for sake of brevity. ** $n$ = 10 in Phases 2/3 because of loss of wheel-running activity. Groups with non-overlapping letters differ significantly from each other by Wilcoxon test

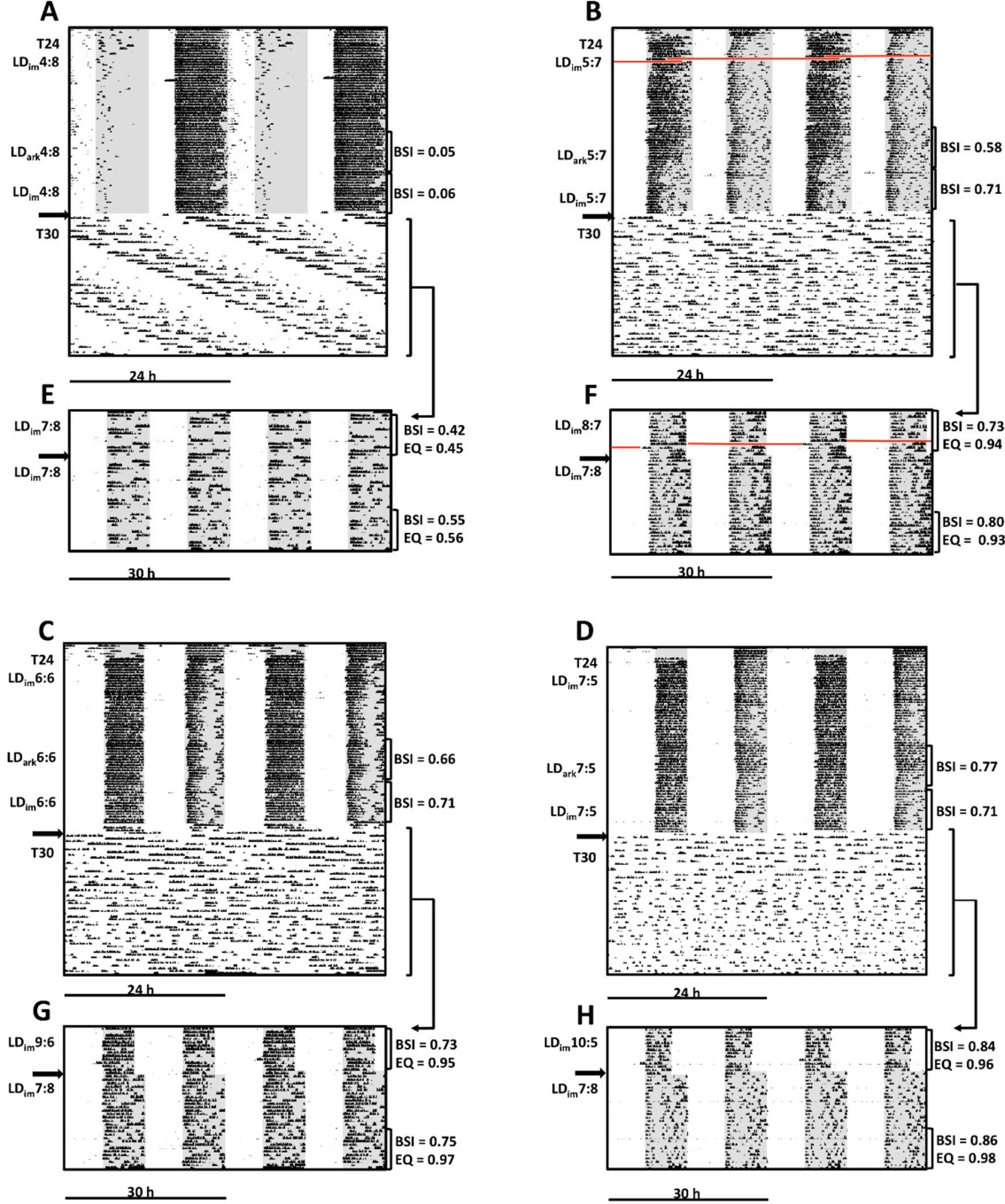

**Figure 3.** Illustrative wheel-running actograms of mice from each condition in Experiment 2. Actograms are double-plotted modulo 24 h (**A**–**D**) or modulo 30 h (**E**–**H**). Mice were initially exposed to $LD_{im}4:8$ $LD_{im}5:7$ $LD_{im}6:6$ or $LD_{im}7:5$ for 8 weeks except that nights were completely dark in weeks 5–6 (Phase 2). Conditions were changed to $LD_{im}7:8$, $LD_{im}8:7$, $LD_{im}9:6$ or $LD_{im}10:5$ for 2 weeks before all were exposed to $LD_{im}7:8$ for 4 additional weeks (Phase 3). For clarity, shading is omitted during T15 $LD_{im}$ cycles in modulo 24 h plots. Other conventions as described in Figure 1. The red line indicates missing data as a result of a technical failure.

Activity rhythms of the same illustrative animals under 15 h $LD_{im}$ cycles are double-plotted both modulo 24 h (lower portions of Figure 3A–D) and modulo 30 h (Figure 3E–H) to facilitate visualization of free-running versus entrained status. The representative mouse in $LD_{im}7:8$ following $LD_{im}4:8$ exhibits a long period, apparently free-running, rhythm (Figure 3A) that also appears negatively

masked by bright light (Figure 3E). EQ values near 0.5 indicate that periodogram power in the circadian range was comparable in magnitude to that at the zeitgeber period (15 or 30 h). Because the rhythm was free-running, activity was sometimes divided between scotophases, which yielded a BSI value also near 0.5 (Figure 3E). In contrast, free-running rhythmicity was not evident in the modulo 24 h plots for each of the mice selected for illustration in other groups ($LD_{im}$8:7, $LD_{im}$9:6 and $LD_{im}$10:5; Figure 3B–D, respectively). Instead, activity appears to be approximately uniformly distributed across all phases of the 24 h day (just as the scotophases are distributed). In modulo 30 h plots (Figure 3F–H), moreover, activity appears well aligned with the scotophases and lacks any salient beating pattern that was evident in Figure 3A. EQ values near 1.0 reflect a near absence of periodogram power at any period other than 15 h. The distribution of activity within each scotophase also appears fairly symmetrically divided, which is reflected in BSI values >0.7. These patterns and high BSI and EQ values persisted in the final experimental phase when these illustrative mice from each group were exposed to $LD_{im}$7:8 (Phase 3).

Analyzing group data, entrainment parameters under T15 differed significantly between groups. In Phase 2, EQ values were lowest in $LD_{im}$7:8 and greatest in $LD_{im}$10:5, with intermediate values for other conditions (Table 3; Figure 4). This pattern reflected significant group differences in periodogram power in the circadian range (23–26 h) rather than at 15 h or 30 h (Table 3). The percentage of activity in the light, a measure reported in earlier experiments to be elevated in animals poorly entrained to T15 [21], was generally low and did not differ here between groups. BSI values also differed by group, but must be interpreted cautiously in light of EQ scores, as poor entrainment (with low EQs) will inflate this index. Finally, in the final weeks when all mice were held under identical $LD_{im}$7:8 (Phase 3), the photic history produced enduring effects on EQ (Table 3; Figure 4). Specifically, mice transferred successively from $LD_{im}$7:5 to $LD_{im}$10:5 to $LD_{im}$7:8 had significantly higher EQ values than mice moved from $LD_{im}$4:8 to $LD_{im}$7:8. Again, the EQ scores appear to be driven principally by periodogram power in the circadian range, which was higher in mice previously exposed to $LD_{im}$4:8/$LD_{im}$7:8 compared to $LD_{im}$7:5/$LD_{im}$10:5. BSI and percentage of activity in light did not differ by group in Phase 3 (Table 3).

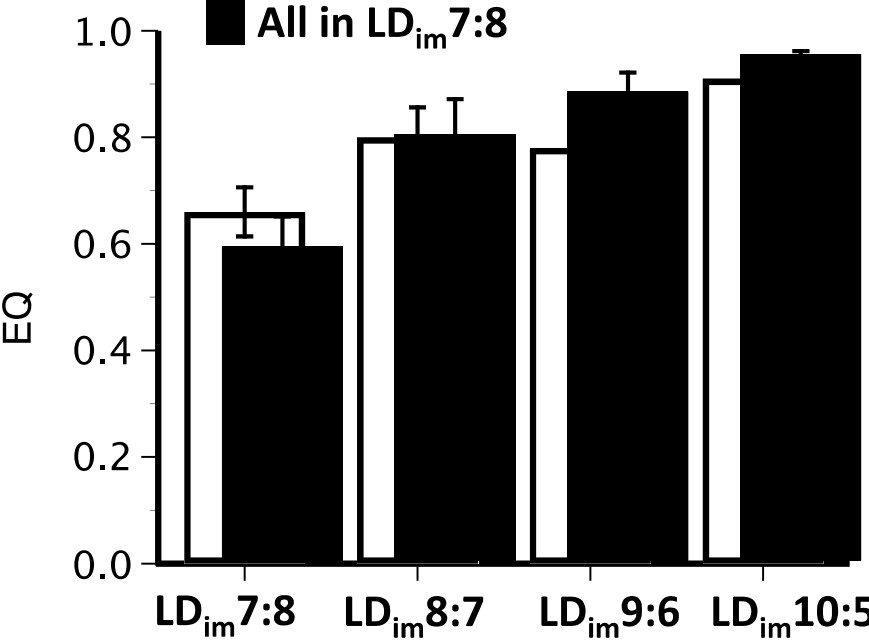

**Figure 4.** Mean ± sem EQ values in T15 $LD_{im}$ conditions of Phase 2 of Experiment 2 that differed in photoperiod (open bars) and in Phase 3 when all animals were maintained in identical $LD_{im}$7:8 (filled bars).

## 4. Discussion

These two experiments replicate and extend findings that rest/activity rhythms of mice are capable of extraordinary entrainment to multi-modal and non 24-h lighting regimes without pharmacological, neurological or genetic intervention and define photoperiodic requirements for inducing and maintaining enhanced entrainment capabilities. Specifically, we confirm that mice exhibit reliable rhythm bifurcation in permissive 24 h $LD_{im}LD_{im}$ conditions and that a history of bifurcation permits subsequent behavioral entrainment to T15 $LD_{im}$. These experiments additionally demonstrate for the first time that induction of rhythm bifurcation in $LD_{im}LD_{im}$ requires exposure to photophases >4 h; that maintenance of bifurcation has different photoperiod requirements than its induction; that behavioral entrainment is possible under a variety of different T15 $LD_{im}$ photoperiods; and that successful entrainment to $LD_{im}7:8$ is an enduring after-effect of remote entrainment conditions. In documenting the extraordinary flexibility of behavioral entrainment in male mice, these results further raise the possibility that circadian systems of other mammalian species, including humans, may be non-invasively rendered more functionally plastic with potential application for shift-work.

Rhythm bifurcation, characterized chiefly by the robust and stable expression of wheel-running activity in two scotophases per 24 h, has been previously described in two hamster species and in mice [20]. Bifurcation (initially referred to as behavioral decoupling or LDLD-induced splitting) was first discovered and examined from an entrainment perspective in Syrian hamsters [28–31]. In that species, shortening of the subject night was hypothesized to catalyze bifurcation based on two observations. First, repeated daytime exposure to novel running wheels led to a progressive shortening of activity duration at night such that a stable bimodal entrainment pattern could be eventually established in the home-cage under LDLD conditions [29]. Second, skeleton photoperiods induced phase-jumps of activity at values approximating conditions under which bifurcated entrainment had been previously observed [30]. We proposed that bifurcation represents a more stable oscillator configuration than a highly compressed subjective night [30]. In both of the current experiments, which manipulated photoperiod directly, longer photophases/shorter scotophases increased BSI, a simple metric of the distribution of activity between two anti-phase scotophases. The response to photoperiod length appears to be graded rather than categorical, as seen by intermediacy of BSI values in $LD_{im}5:7$ and $LD_{im}6:6$. Statistically, this intermediacy was confirmed for 24 h periodogram power in Phase 1 of each experiment, even if the groups were not statistically distinct in every circadian measure. The response to photoperiod was not monotonic as lengthening photophases from 8 h to 10 h in Experiment 1 reduced BSI values. Visual inspection of actograms also revealed free-running activity components and generally less stable activity patterns in this condition. Collectively, these experiments directly establish a photoperiod dependence of rhythm bifurcation and suggest that $LD_{im}7:5$ and $LD_{im}8:4$ may approximate optimal conditions for this phenomenon in our laboratory. Of course, such conditions would never be encountered under natural conditions, and optimality is defined here in terms of a mechanistic rather than an adaptive perspective.

If, as previously suggested [30], bifurcation entrainment represents a more stable condition than unbifurcated entrainment to particular photoperiods (e.g., $LD_{im}8:4$), it raises the possibility that the circadian system may be bistable under certain photoperiods. Hence, we tested whether the photoperiod requirements to maintain bifurcation would differ from those to induce it. Indeed, mice with different photoperiodic histories maintained group differences in BSI even after 4 weeks of exposure to identical $LD_{im}4:8$ photoperiods. Thus, mice could be reliably rendered bifurcated or unbifurcated under $LD_{im}4:8$ by either exposing them 4 weeks of inductive conditions or not, respectively. The persistence of differentially entrained behavior under identical light conditions underscores the conclusion that bifurcation is a bona fide entrainment condition rather than positive or negative masking by light/dark. Although more work remains to be done on this issue, an altered entrainment configuration of the circadian pacemaker in bifurcation was previously supported by studies of SCN clock gene/protein expression, melatonin secretion, and behavior in hamsters and mice [32–34]. Finally, phase analysis of the free-running rhythm in $D_{im}D_{im}$ provides one further

indicator of the pacemaker-entraining effects of the prior light regimens. In this regard, it is notable that groups with the most extreme BSI values in Phase 1 ($LD_{im}4{:}8$ and $LD_{im}8{:}4$ were lowest and highest, respectively) had activity onsets that were each significantly and similarly clustered, suggesting that these were most effectively controlled by their zeitgebers. In contrast, the scattered phases of activity onsets after $LD_{im}6{:}6$ and $LD_{im}10{:}2$ corroborated the impression that entrainment was less stable or well controlled under these regimens. A stabilizing role of longer photoperiods for bifurcation maintenance is further evidenced by the general decline in BSI values across groups from Phase 1 to Phase 2.

A notable and attractive feature of rhythm bifurcation in 24 h LDLD cycles is that it appears to render circadian systems more flexible in re-entraining to changing phases or zeitgeber periods. Specifically, bifurcated hamsters exhibit accelerated recovery from jetlag [35]; ex vivo rhythms of PER2::LUC expression in the SCN are more strongly reset by dissection in bifurcated mice [36]; and bifurcated but not unbifurcated mice readily synchronize their locomotor activity rhythms to LD10:5 [21]. Proper entrainment to $LD_{im}10{:}5$ was confirmed previously by assessing control of free-running phase under constant conditions [21]. Surprisingly, that study also demonstrated that the distribution of wheel-running activity between alternate scotophases was less symmetric in T24 LDLD than it was in T30 LDLD, raising the intriguing possibility that the latter represented a more strictly light-controlled entrainment condition than the former. Here, we replicate and extend our findings of entrainment to T15/30 conditions by examining its photoperiod-dependence for the first time. EQ values, which capture how well behavioral rhythms match the period of the zeitgeber, increased in T15/30 in a graded fashion as photophases were lengthened, nearly reaching unity under $LD_{im}10{:}5$. As with BSI in T24 in Experiment 1, acute versus entrainment effects of light could be distinguished in Phase 3 exposure to $LD_{im}7{:}8$. Entrainment to $LD_{im}7{:}8$ was clearly superior in animals coming from $LD_{im}7{:}5/LD_{im}10{:}5$ than in those coming from $LD_{im}4{:}8/LD_{im}7{:}8$. The latter was the only group with salient free-running rhythmicity in $LD_{im}7{:}8$ (e.g., Figure 3E). Thus, even after 4 weeks of identical T15/30 conditions, an after-effect of prior photoperiod could be discerned in the entrainability to $LD_{im}7{:}8$. In contrast with BSI values in T24, a clear graded dependence of any entrainment measure (EQ, % activity in light, periodogram power at any value) on prior photoperiod could not be established.

In each of three species studied to date, dim nighttime illumination facilitates bifurcation in LDLD [21,26,37], which is rare under completely dark nights. Just as long photoperiods were not required to maintain elevated BSIs, extinction of dim light after 4 weeks of $LD_{im}LD_{im}$ and its restoration 2 weeks later did not discernibly perturb entrainment to the otherwise unchanging LDLD cycles. Comparably, discontinuation of dim light did not preclude successful entrainment to T15/30 in previously bifurcated mice [21] but did alter bifurcation stability in Syrian hamsters [38]. Moreover, dim scotophase illumination can exert strong effects on its own in contexts not involving bifurcated entrainment (e.g., accelerating re-entrainment to typical LD cycles; arrhythmicity in constant conditions; enhanced seasonal responses etc. [39,40]). Because studies of bifurcation generally require dim light, it is not possible in many experimental designs to definitively distinguish the effects of *bifurcation* on enhanced entrainment from those of dim light per se. The latter cannot account, however, for different entrainment quality in Phase 2 of Experiment 1 and in Phase 3 of Experiment 2 since both involved continuous dim light exposure. We therefore interpret these enduring group differences as the result of altered entrainment status. We cannot, however, exclude the possibility that dim light may additionally be directly facilitating entrainment to these shared conditions. Such an effect would diminish group differences and therefore lead us to underestimate any effects mediated via prior entrainment status. Subjectively, we were frankly surprised by the strong behavioral entrainment in so many of the mice in $LD_{im}7{:}8$ and the less common occurrence of obviously free-running rhythms. An earlier study demonstrated that that mice adapted equally well to $LD_{im}13{:}5$ after a series of complex manipulations involving bifurcated versus unbifurcated histories [22]. A simple follow-up to the latter study demonstrated that either a bifurcated history or dim light without bifurcation was sufficient to permit this T18 entrainment [22]. Additional studies, therefore, are required to determine the contribution, if any, of bifurcation-independent actions of dim light on entrainment to $LD_{im}7{:}8$.

The study of bifurcation and extraordinary entrainment taxes the nomenclature and conventions of chronobiology designed for rhythms as they occur under natural conditions. Historically, two fundamental and related challenges of an emergent chronobiology were to establish the endogenous (i.e., circadian) rather than environmental (i.e., masking) origin of measured rhythms and to demonstrate how robust and stable pacemaker properties ensures proper entrainment under ecologically valid conditions. Accordingly, stringent entrainment criteria have been relied upon, including stable free-running rhythmicity in constant conditions; reproducible phase and period control under a zeitgeber; and predictable relationships between phase response curves, free-running periods and phase angles of entrainment. These exacting criteria are not met with rhythm bifurcation or T15 entrainment. For example, bifurcated rhythms quickly reorganize into unimodal rhythms under constant conditions. Phase angles of entrainment do not conform to non-parametric entrainment theory [23], and strong period after-effects are not apparent in constant dark [21]. These differences notwithstanding, the extraordinary behavioral adaptations observed here and in related papers [21, 22,28] are inconsistent with traditional accounts of masking, in which light and dark may modify rhythms by addition or subtraction but which otherwise remain coupled to an endogenous oscillator. Findings that discount simple masking accounts include entrainment after-effects (present study), phase control in DD [21], altered phase-resetting [35,36], and reorganization of SCN and melatonin rhythmicity [32–34]. While a simple masking interpretation can be rejected, the mechanistic basis of the enhanced behavioral entrainment remains to be determined. In computational models, reduction of pacemaker amplitude or changes in coupling, for example, can increase the range of their entrainment and other manifestations of circadian plasticity that overlap findings reported here [41–43]. Disruption of molecular mechanisms that putatively mediate neuronal coupling in the SCN likewise can result in enhanced behavioral flexibility [42,44]. Whether these or alternative mechanism are relevant to bifurcation awaits further investigation.

## 5. Conclusions

The use of atypical lighting conditions such as dim light at night and polyphasic light:dark conditions reveals extreme flexibility in the entrainment of rodent rest/activity rhythms. Rhythm bifurcation in 24 h LDLD cycles is facilitated by exposure to longer photophases and shorter scotophases, but maintenance of bifurcation is less strictly dependent on photoperiod. Bifurcated mice adapt rhythms to match 30 h LDLD cycles, again with differential photoperiod dependence for induction and maintenance. These enduring entrainment after-effects await mechanistic explanation and exploration of translational potential.

**Author Contributions:** Conceptualization, J.S. and M.R.G.; Data curation, J.S., A.H.F. and D.A.M.J.; Formal analysis, J.S., A.H.F., D.A.M.J. and M.R.G.; Funding acquisition, M.R.G.; Investigation, J.S.; Methodology, J.S., A.H.F., D.A.M.J. and M.R.G.; Supervision, M.R.G.; Visualization, J.S. and A.H.F.; Writing—original draft, J.S. and M.R.G.; Writing—review & editing, J.S., A.H.F., D.A.M.J. and M.R.G.

**Funding:** This work was funded by the US Office of Naval Research grant N00014-13-1-0285.

**Acknowledgments:** The authors thank the animal husbandry staff for their outstanding animal care and Thijs Walbeek for comments on an earlier draft of the manuscript.

**Conflicts of Interest:** The authors declare no conflict of interest. The funders had no role in the design of the study; in the collection, analyses, or interpretation of data; in the writing of the manuscript, or in the decision to publish the results.

## Abbreviations

| | |
|---|---|
| LDLD | light:dark:light:dark |
| SCN | suprachiasmatic nuclei |
| T12 | 12 h cycle length |
| BSI | Bifurcation symmetry index |
| EQ | Entrainment quotient |
| $LD_{im}$ | Light:dim |

$D_{im}D_{im}$    Constant dim light
PRC      Phase response curve
FRP      Free-running period

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
