# Peer review of "Photoperiodic Requirements for Induction and Maintenance of Rhythm Bifurcation and Extraordinary Entrainment in Male Mice"

_2624-5175, doi:10.3390/clockssleep1030025_

Round 1

Reviewer 1 Report

The authors study the entrainment of activity rhythms in mice to a variety of light conditions (dim light, double waveforms, short LD cycles…). They reproduce previous findings on bifurcations and entrainment to extreme LD cycles. As novel results they explore the role of photoperiod and the combination of pretreatment with bifurcations and extreme cycles afterwards. They find a surprising plasticity to varying protocols with potential relevance to shift work optimization.

The results are clear, the statistical analysis is carefully done, and the conclusions are justified. However, the manuscript could be improved as discussed below.

Specific comments:

1.   The authors implicitely assume that the readers are familiar with the underlying LDLD protocol and bifurcations. I suggest a better motivation and clarification of terminology in the abstract and introduction. Why LDLD protocols are important in chronobiology? Is there any connection of their “bifurcation” to other bifurcations (aorta, nonlinear dynamics terminology)?

2.   The references are heavily biased. Photoperiods, entrainment, LD cycles, seasonality, shift work have been studied my many groups including Rosbash, Merrow, Okamura, Piggins, Myung, Yoshimura, Erzberger, Hastings. How LDLD protocols complement these studies?

3.   What is the omnibus effect? Perhaps the statement can be reformulated with variances.

4.   The discussion is a bit like a review of previous Gorman papers. It might be focused a bit.

5.   I find the statement: “Phase angles of entrainment do not conform …” quite strong. The theory of entrainment phase is quite complex (see e.g. Aschoff/Pohl 1978, Wever, Granada 2013, Schmal 2015) and thus some advanced theory including amplitudes and 2:1 entrainment might be closer to the data.

6.   The fast re-entrainment of bifurcated rodents reminds me to the theory of weak and strong oscillators. For reduced amplitudes and synchronization many studies report faster entrainment (Abraham 2010, Okamura and AVP receptor knockouts, Herzog and VIP injection). Thus the LDLD protocol seems to represent another way to weakening the circadian oscillator.

Author Response

General response.  The reviewer raises many excellent points about the broader contextualization of our project.  We are grateful for the mention of specific references, some of which we were aware and others that were new to us. The comments are guiding our writing of manuscripts that we are currently drafting.  We have elected not to attempt a substantial rewriting of the paper to address the very useful, but general, comments in part because of the editor's request for a rapid turn-around. Nevertheless, we are extremely thankful to the reviewer for the comments, which we take to heart .

1.   The authors implicitly assume that the readers are familiar with the underlying LDLD protocol and bifurcations. I suggest a better motivation and clarification of terminology in the abstract and introduction. Why LDLD protocols are important in chronobiology? Is there any connection of their “bifurcation” to other bifurcations (aorta, nonlinear dynamics terminology)?

Response.  We have made modest revisions to the abstract and introduction to provide additional context and clarify our use of the term "bifurcation."

2.   The references are heavily biased. Photoperiods, entrainment, LD cycles, seasonality, shift work have been studied my many groups including Rosbash, Merrow, Okamura, Piggins, Myung, Yoshimura, Erzberger, Hastings. How LDLD protocols complement these studies?

Response: We acknowledge the important contributions of these (and other labs). We have added some additional references to provide general context and where closely related to the details of the present project.

3.   What is the omnibus effect? Perhaps the statement can be reformulated with variances.

Response: We rephrased our characterization of our statistical testing workflow.

4.   The discussion is a bit like a review of previous Gorman papers. It might be focused a bit.

Response: We accept the criticism, even if we are uncertain as to how we might address it in the present manuscript.  Historically, we have struggled with balancing concerns related to unfamiliarity of most readers with bifurcation (we are the only lab that focuses on this) and avoiding regurgitation of our own papers. It is probably in our best interest to write a concise review that puts the most critical issues in a single place. 

5.   I find the statement: “Phase angles of entrainment do not conform …” quite strong. The theory of entrainment phase is quite complex (see e.g. Aschoff/Pohl 1978, Wever, Granada 2013, Schmal 2015) and thus some advanced theory including amplitudes and 2:1 entrainment might be closer to the data.

Response. We agree and have modified our claim to be more precise.

6.   The fast re-entrainment of bifurcated rodents reminds me to the theory of weak and strong oscillators. For reduced amplitudes and synchronization many studies report faster entrainment (Abraham 2010, Okamura and AVP receptor knockouts, Herzog and VIP injection). Thus the LDLD protocol seems to represent another way to weakening the circadian oscillator.

Response: We agree and have added a few sentences at the end of the discussion that highlights this possibility.

Reviewer 2 Report

This is a beautifully constructed manuscript and elegant study by Sun and colleagues. It investigates the lighting schedules required to induce and maintain bifurcation, and enhancement of the range of apparent entrainment. This study builds on previous work from this lab. It is timely, and advances our knowledge of the circadian clock, with the potential to eventually help sections of the society with circadian disruption.

The experiments are well-designed, robustly executed and analysed, and the results are clearly reported. The manuscript is succinctly written, and I have only some minor comments for the Author’s consideration.

1)      Please provide one or two review references for the sentence in lines: 43-45.

2)      After each of the lighting conditions (for example, figure 1B, T24, LD6:6, 4:8), would it be informative to liberate the mice into DD, total darkness not Dim, to assess their endogenous periods, and later compare the after-effects – e.g. DD1, DD2 etc?

3)      Where appropriate, it would be helpful if the authors could provide running wheel activity mean waveforms.

4)      Page 8: Lines 249-250: Sentence beginning with “Rather….”: Could the authors clarify this statement further? As is written, it is a bit difficult to understand.

5)      In the legend of figure 3, third line, please indicate the corresponding panel (A, B or C etc…) after each LD cycle, as shown in the figure 1 legend.

6)      There are red lines on panels B and F of figure 3. Could the authors indicate what these lines refer to?

7)      There is a small mistake in line 362 (studies “in”?) SCN…..

It will be fantastic to try this in a diurnal species.

Author Response

This is a beautifully constructed manuscript and elegant study by Sun and colleagues. It investigates the lighting schedules required to induce and maintain bifurcation, and enhancement of the range of apparent entrainment. This study builds on previous work from this lab. It is timely, and advances our knowledge of the circadian clock, with the potential to eventually help sections of the society with circadian disruption.

The experiments are well-designed, robustly executed and analysed, and the results are clearly reported. The manuscript is succinctly written, and I have only some minor comments for the Author’s consideration.

Response: Thank you for the very kind evaluation.

1)      Please provide one or two review references for the sentence in lines: 43-45.

Response:  We added a recent reference as suggested.

2)      After each of the lighting conditions (for example, figure 1B, T24, LD6:6, 4:8), would it be informative to liberate the mice into DD, total darkness not Dim, to assess their endogenous periods, and later compare the after-effects – e.g. DD1, DD2 etc?

Response: We agree that more data from constant conditions (dark included) would be interesting.  However, we view such exposure as a one-way trip. Because of potential importance of photoperiodic history, we try to avoid re-entraining animals after exposure to constant dark.  Thus, we don't have the data that the reviewer mentions.

3)     Where appropriate, it would be helpful if the authors could provide running wheel activity mean waveforms.

Response: We tried adding waveforms to the representative actogram figures calculated over the same intervals as BSI/EQ.  In our judgment, it cluttered the figure and didn't offer much insight.  Additionally, we hesitated to calculate waveforms where there was poor entrainment because the resultant curve is not representative of any single day of activity. Establishing an entrainment threshold for inclusion introduced more complexity and subjectively than we thought justified.  We therefore elected not to incorporate waveforms into the presentation.

4)      Page 8: Lines 249-250: Sentence beginning with “Rather….”: Could the authors clarify this statement further? As is written, it is a bit difficult to understand.

Response: We rephrased this sentence.

5)      In the legend of figure 3, third line, please indicate the corresponding panel (A, B or C etc…) after each LD cycle, as shown in the figure 1 legend.

Response: We corrected this.

6)      There are red lines on panels B and F of figure 3. Could the authors indicate what these lines refer to?

Response: We added a note in the legend that this is an interval of missing data.

7)      There is a small mistake in line 362 (studies “in”?) SCN…..

Response:  This has been corrected.